

# Reduced floating-point precision in regional climate simulations: An ensemble-based statistical verification

Banderier Hugo[1, 2], Zeman Christian[1], Leutwyler David[1, 3], Rüdisühli Stefan[1], and Schär Christoph[1]

[1]Institute for Atmospheric and Climate Science, ETH Zürich, Switzerland
[2]Current affiliation : Oeschger Center for Climate Change Research and Geography Institute, Universität Bern, Switzerland
[3]Current affiliation : Federal Institute of Meteorology and Climatology, MeteoSwiss, Zurich, Switzerland

**Correspondence:** Banderier Hugo (hugo.banderier@unibe.ch)

**Abstract.** The use of single precision in floating-point representation has become increasingly common in operational weather prediction. Meanwhile, climate simulations are still typically run in double precision. The reasons for this are likely manifold and range from concerns about compliance to conservation laws to the unknown effect of single precision on slow processes, or simply the less frequent opportunity and higher computational costs of validation.

Using an ensemble-based statistical methodology, Zeman and Schär (2022) could detect differences between double- and single-precision simulations from the regional weather and climate model COSMO. However, these differences are minimal and often only detectable during the first few hours or days of the simulation. To evaluate whether these differences are relevant for regional climate simulations, we have conducted 10-year-long ensemble simulations over the EURO-CORDEX domain in single and double precision with 100 ensemble members.

By applying the statistical testing at a grid-cell level for 47 output variables every 12 or 24 hours, we only detected a marginally increased rejection rate for the single-precision climate simulations compared to the double-precision reference. This increase in the rejection rate is much smaller than that arising from minor variations of the horizontal diffusion coefficient in the model. Therefore, we deem it negligible.

To our knowledge, this study represents the most comprehensive analysis so far on the effects of reduced precision in a 15  climate simulation for a realistic setting, namely with a fully-fledged regional climate model in a configuration that has already been used for climate change impact and adaptation studies. The ensemble-based verification of model output at a grid-cell level and high temporal resolution is very sensitive and suitable for verifying climate models. Furthermore, the verification methodology is model agnostic, meaning it can be applied to any model. Our findings encourage exploiting the reduction of computational costs ( $\sim 30\%$ for COSMO) obtained from reduced precision for regional climate simulations.

## 1  Introduction


Numerical Weather and Climate models have evolved from simple and computationally inexpensive radiative-convective models (Manabe and Wetherald, 1967) into highly complex codes solving the governing equations on billions of grid points. While this advancement improves the representation of Earth's climate, it also escalates computational costs and energy consump-



tion (Schär et al., 2020). Reducing these costs without compromising model accuracy is crucial for further enhancements of
resolution, domain size, and ensemble size.

Reducing the precision in floating-point representation is a straightforward method to alleviate computational costs in nu-
merical simulations. Typically, a floating-point number in weather and climate models requires 64 bits (double precision; DP).
Reducing the precision to 32 bits (single precision; SP) will reduce the dynamical range ($\pm 10^{308}$ for DP, $\pm 10^{38}$ for SP) and the
accuracy (machine precision; $\sim 10^{-16}$ for DP, $\sim 10^{-7}$ for SP) of numbers. Take the representation of temperature for instance.
In double precision, temperature can be represented with a very high level of detail, for example, 296.45678912345678 K. In
single precision, the number of digits could be reduced to something like 296.456789 K. For most calculations within a climate
model, the range and accuracy of single precision are more than enough, especially when considering the often substantial
uncertainties associated with discretization, physical parameterizations for subgrid-scale processes, and emission scenarios.
On the plus side, the use of single precision instead of double precision can significantly reduce the computational costs of a
simulation due to higher operational intensity (number of floating-point operations per number of bytes transferred between
cache and shared memory) and a reduction of internode communication, as single precision allows for fitting twice as many
grid points into the memory of a single node compared to double precision.

Reduced precision is used operationally by MeteoSwiss for the national weather forecast for Switzerland. They implemented
single precision for most model components, observing no discernible impact on forecast skill (Rüdisühli et al., 2014). While
the switch to single precision did not require any major code changes, some modifications like the reformulation of specific
formulas and the addition of precision-dependent epsilons were necessary for the comparison of floating-point numbers (i.e.,
`ABS(a-b) < eps`) and to avoid division by zero (see Rüdisühli et al., 2014, for an overview). A few model components still
rely on double precision, namely parts of the radiation and the soil model, making the implementation "mixed precision" rather
than purely SP. The reduction in computational costs from using COSMO with reduced precision is likely highly dependent
on the model domain and domain decomposition, but it can expected to be around 30% (Zeman and Schär, 2022) or even 40%
(Rüdisühli et al., 2014).

The Unified Model (Brown et al., 2012) from the Met Office operationally uses single precision in the iterative solver for
the Helmholtz equation in the dynamical core, which leads to improved runtime with no detrimental impact on the accuracy of
the solution (Maynard and Walters, 2019).
Düben and Palmer (2014) performed global simulations with the European Centre for Medium-Range Weather Forecasts
(ECMWF) Integrated Forecasting System (IFS; ECMWF, 2023) at single precision for several days at horizontal grid spacings
ranging from 950 km to 32 km. The differences in 500 hPa geopotential and 850 hPa temperature between SP and DP were
consistently smaller than those between different DP ensemble members of the standard ensemble forecasting system.

Váňa et al. (2017) performed 13-months-long global simulations at $\sim 50$ km grid spacing with four ensemble members
(generated by shifting the initial times) using the IFS in DP and SP, with the SP runs having around 40% shorter runtimes.
They compared the results with observations and found only minor differences in root-mean-square errors (RMSE) from annual
means of several model quantities between the DP and SP versions. These differences were negligible when considering the
magnitude of systematic forecast errors.



Nakano et al. (2018) evaluated SP for parts of the Nonhydrostatic Icosahedral Grid Atmospheric Model (NICAM; Satoh et al., 2014) using the Jablonowski and Williamson baroclinic wave benchmark test (Jablonowski and Williamson, 2006). By using DP for the model setup and SP everywhere else, the simulations showed the same quality as those with the conventional DP model but could be performed 1.8 times faster.

Klöwer et al. (2020) investigated the effect of 16-bit precision on a shallow water model and found that without mitigation methods such as rescaling, reordering, or using higher precision for critical parts of the code, 16-bit arithmetic induced too big rounding errors. They also showed that using the Posit number format (Gustafson and Yonemoto, 2017) reduced the forecast error compared to the traditional IEEE 754 format. Ackmann et al. (2022) performed idealized tests with a shallow water model and showed that at least some parts of the elliptic solvers of the semi-implicit time-stepping schemes could be performed at half precision without negatively affecting the quality of the solution when compared to only using double precision.

Extensive testing with reduced precision for IFS was conducted by Lang et al. (2021). They performed medium-range ensembles consisting of 50 members with a forecast range of 15 days for several periods at a grid spacing of 18 km and 91 or 137 vertical levels. The simulations were validated using the continuous ranked probability score (CRPS; Matheson and Winkler, 1976) against model analysis and observations for 121 variables, and significance has been tested with the Student's t-test combined with variance inflation accounting for temporal autocorrelation following the approach by Geer (2015). Compared to the DP ensemble with 91 levels, the SP ensemble with 91 levels had a reduced runtime by approximately 40% (same as Rüdisühli et al., 2014 and Váňa et al., 2017) without compromising forecast skill. Moreover, the SP ensemble with 137 levels significantly improved forecast skill while still having about 10% shorter runtime than the DP ensemble with 91 levels. Based on these findings and previous studies, ECWMF adopted SP for its ensemble and deterministic forecasts starting with IFS model cycle 47R2 (Lang et al., 2021).

While single or mixed precision is becoming increasingly common for weather forecasts, most climate simulations are still being run in double precision. There are several good reasons to err on the side of caution with climate simulations. Weather forecasts are performed many times every day and can thus be validated often and promptly by comparing them to observations. Climate simulations are performed much less frequently, and their validation is much more involved and less routine. Furthermore, the slow processes, such as the ocean model, the soil model, or the representation of ice sheets, play a significantly more important role, as well as compliance with conservation laws. Thus, demonstrating no degradation in weather forecast skill with reduced precision in an atmospheric model does not automatically guarantee the same outcome for climate simulations.

One of the few studies on the effect of reduced precision on climate timescales was conducted by Chantry et al. (2019). They performed global 10-member ensemble simulations for 10 years at ∼ 125 km grid spacing with double and reduced precision. To assess the differences, they applied a grid-point-based Student's t-test combined with the false discovery rate (FDR) test on decadal averages of precipitation, 2 m temperature, and surface pressure. Their findings indicated no significant differences between DP and SP in the measured fields. Interestingly, no statistically significant differences were found even when using half precision. However, in this case, the zeroth mode of the spectral part of the model, which represents the global mean of a field, was retained in DP to mitigate large roundoff errors for quantities like geopotential or temperature.



In their study, Paxton et al. (2022) evaluated ensembles with different floating-point precisions using the modified SPEEDY model (Molteni, 2003; Kucharski et al., 2006, 2013; Saffin et al., 2020). They integrated five members per ensemble for 10 years and calculated the Wasserstein distance for various variables on both grid points and the entire grid. By reducing the number of significand bits while keeping the exponent bits constant, they found negligible model differences for geopotential height, horizontal wind speed, and precipitation when using 14 significand bits instead of 53. They also highlighted the benefits of stochastic rounding in mitigating errors induced by reduced precision.

Similarly, Kimpson et al. (2023) examined the effects of reduced precision on climate change simulations using the SPEEDY model. They ran ensembles with five members for 100 years, focusing on increased $CO_2$ concentrations. Compared to a DP ensemble, the ensembles with reduced precision accurately represented global mean surface temperature and precipitation. Notably, even an ensemble with 10 significand bits showed biases within $\sim 0.1$K for temperature and $\sim 0.015$mm$(6$h$)^{-1}$ for precipitation. Similar to Paxton et al. (2022), they also found that stochastic rounding reduced global mean biases.

The recent studies conducted by Chantry et al. (2019), Paxton et al. (2022), and Kimpson et al. (2023) provide strong motivation for employing reduced precision in climate simulations. However, some may still harbor reservations due to the emphasis on decadal averages of select variables (Chantry et al., 2019) or the utilization of simplified parameterizations (Paxton et al., 2022; Kimpson et al., 2023). These considerations may raise doubts and hinder full commitment to adopting reduced precision techniques in climate simulations.

In this study, we aim to systematically assess the effect of reduced precision using the ensemble-based verification methodology by Zeman and Schär (2022). This methodology was originally designed to detect small changes in model behavior resulting from hardware infrastructure changes or software updates. The methodology offers high sensitivity and employs statistical testing at the grid-cell level for instantaneous, hourly, or daily output variables.

The methodology is applied to 10-year-long regional climate model ensemble simulations, consisting of 100 members per ensemble, with the COSMO-crCLIM model on the European domain of the Coordinated Regional Climate Downscaling Experiment (CORDEX). The simulations employ a horizontal grid spacing of $0.44°$ ($\sim 50$ km) and are configured identically to the model in its contribution to CORDEX-EU (see Sørland et al., 2021, for more information). By applying this verification approach to all relevant output fields, including those of the CORDEX ensemble, we aim to provide a comprehensive assessment of the differences caused by the switch from DP to SP. The results are then compared to ensemble simulations with slightly increased horizontal diffusion to better quantify the sensitivity of the methodology to small model modifications. The methodology is compared to the popular Benjamini-Hochberg procedure in Appendix A, while Appendix B explores the effects of a coarser time resolution on the methodology's sensitivity and Appendix C explores a small technical caveat of the methodology in edge cases, and its solution.

## 2 Methods and data

### 2.1 Statistical methodology

The methodology used in this work was developed in Zeman and Schär (2022) and is briefly described here.



We consider ensemble simulations from two versions of a climate model: an 'old' model and a 'new' model. The methodology aims to to assert whether or not the results from the two versions can be statistically distinguished from each other. In the context of statistical hypothesis testing, we define a global null hypothesis for each tested output variable ($\varphi$) and model time step. The global null hypothesis is as follows:

– The results from the old and the new model, $\varphi_{\mathrm{old}}$ and $\varphi_{\mathrm{new}}$, are drawn from the same distribution.

We consider the versions of the model significantly different at this output time step if we reject the global hypothesis. Because two field distributions cannot be compared directly, we perform local hypothesis testing on a grid point level, evaluating each grid point individually. To this end, we use the two-sample Kolmogorov-Smirnov (KS) test on the distribution of $\varphi$ at each grid point. These local tests have the following null hypotheses:

– The results from the old and the new model at grid point $(i,j)$, are drawn from the same distribution.

Deciding whether or not to reject a global null hypothesis based on the rejection results of local null hypotheses poses a ubiquitous problem in climate sciences. This issue is discussed more thoroughly in, e.g., Zeman and Schär (2022); Wilks (2016). Our methodology addresses this problem by employing a combination of Monte Carlo methods, subsampling, and a control ensemble. A schematic overview of the methodology is provided in Fig. 1.

The reference and control ensemble are both generated with the old model and the test ensemble with the new model. Each ensemble comprises $n_{\mathrm{mem}} = 100$ members. To perform the analysis, we employ subsampling by conducting $n_{\mathrm{sel}} = 100$ subsamples, consisting of randomly drawn $n_{\mathrm{sam}} = 75$ members. For each subsample, we test the local null hypothesis at each grid point between the reference (R) and control (C) ensembles, as well as between the reference (R) and test (T) ensembles. The outcomes of these tests (whether they are rejected or not) are then spatially averaged to obtain a rejection rate for each pair of tests. By repeating this process (subsampling — local testing — spatial averaging) $n_{\mathrm{sel}}$ times, we obtain two empirical distribution functions of rejection rates, denoted as $\hat{f}_{C-R}$ and $\hat{f}_{T-R}$, which correspond to the control–reference and test–reference pairs, respectively.

Finally, we reject the global null hypothesis if the mean of $\hat{f}_{T-R}$ exceeds the $95^{\mathrm{th}}$ percentile of $\hat{f}_{C-R}$. In other words, when the test ensemble, generated by the new model, exhibits a substantially higher number of local rejections when compared to the reference ensemble, it indicates that the new model differs from the old model on a global scale.

### 2.2 Data

The methodology is applied to daily and 12-hourly output from 10-year-long regional climate simulations with the COSMO 6.0 model. These simulations are conducted on hybrid GPU-CPU nodes on the supercomputer Piz Daint operated by the Swiss National Supercomputing Centre (CSCS). The experimental setup follows the configuration of the EURO-Cordex EUR44 experiments (Sørland et al., 2021), which employ a lon-lat grid consisting of $129 \times 132$ points over a European domain with a rotated pole to ensure uniform grid spacing of $0.44°$ ($\sim 50\,\mathrm{km}$). The simulations are driven by boundary conditions derived from the ERA-Interim reanalysis (Dee et al., 2011). The outermost 10 grid points, identified as the nudging zone, are not considered in the analysis.





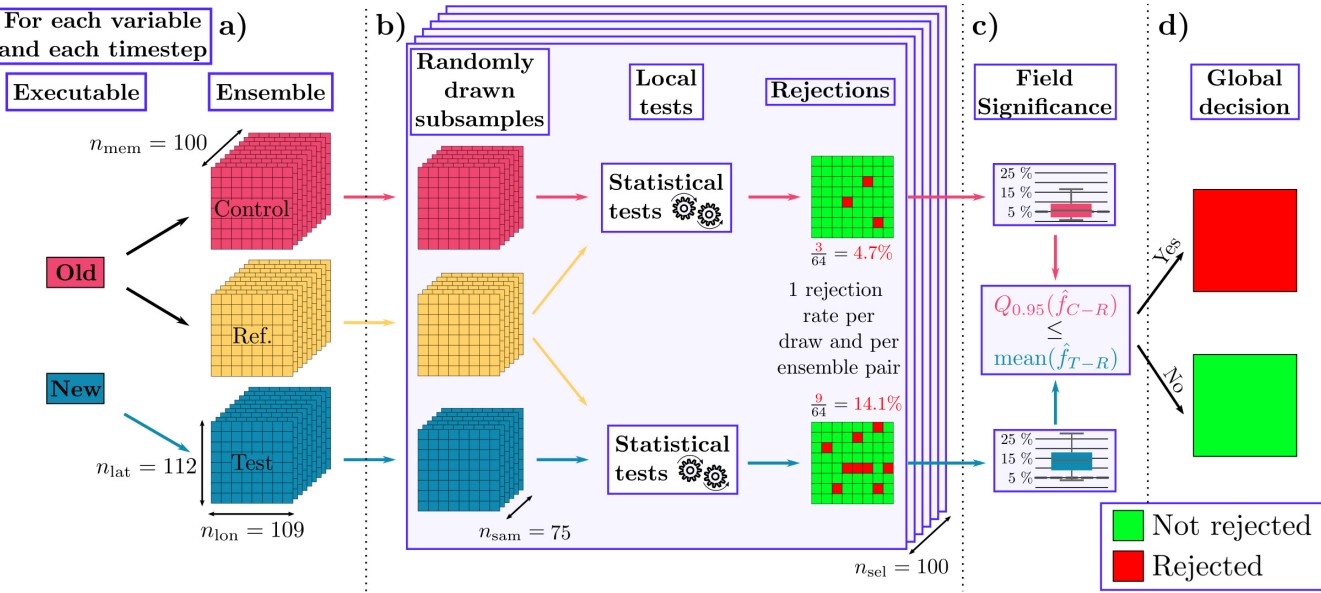

**Figure 1.** Workflow of the statistical test. (Panel a) Control and reference ensembles are produced by an old model and a test ensemble is produced by a new model. (Panel b) Subsequently, subsamples from each of the ensembles are drawn, and a statistical test is performed at each grid point of each subsample. The two resulting arrays of local rejections, are then averaged into a global rejection rate. (Panel c) The rejection rates from each subsampling step then form the two empirical distribution functions $\hat{f}_{C-R}$ and $\hat{f}_{T-R}$. We compare the $95^{\text{th}}$ percentile of $\hat{f}_{C-R}$ against the mean $\hat{f}_{T-R}$. (Panel d) If the mean is larger, the global null hypothesis is rejected.

In total, seven ensemble simulations of 100 members each are performed. The ensembles are created by adding random perturbations to the initial conditions of seven prognostic variables: the three wind components, pressure, temperature, specific humidity, and cloud water content. The perturbations take the form $\varphi_p = (1 + \epsilon R)\varphi$ where $\varphi_p$ and $\varphi$ respectively represent the perturbed and unperturbed variables, $R$ denotes a random number between $-1$ and $1$, and $\epsilon$ is a small value (here $\epsilon = 10^{-4}$). Only initial conditions are perturbed. The lateral and upper boundary conditions are the same across all members and ensembles. This expensive total of 7000 simulated years is only necessary to assess the performance of our methodology, and only feasible thanks to the relatively low grid resolution we use. The discussion section provides readers recommendations to check the performance of their SP implementation in a far cheaper way.

Out of the seven ensembles, three are created with parameters considered as "default". Two of these ensembles serve as the control and reference ensembles required by the methodology described in the previous section, while the third constitutes an identical test ensemble where every rejection is essentially a false positive. A fourth ensemble uses SP instead of the default DP and constitutes the main test. Finally, the sensitivity and performance of the methodology are evaluated using three additional ensembles using DP floats but different horizontal diffusion coefficients. These diffusion coefficients correspond to values of $C = 0.33$, $C = 0.41$, and $C = 0.50$, whereas the default value is $C = 0.25$. While increased diffusion will have a significant





effect on the model behavior (Zeman et al., 2021), the magnitude of the changes above is small and lies within the range used
for model tuning (Schättler et al., 2021).

## 3 Results

Our results consist of a Boolean decision time series for each of the 47 variables tested across five different test ensembles: the
Identity Test (ID), the Reduced Precision Test (SP), and the Modified Diffusion Tests with varying coefficients (C33, C41, and
C50). These time series, with resolutions corresponding to the output variables, indicate whether the test was rejected (true) or
passed (false).

A visual representation of our results is provided in Fig. 2, displaying monthly averaged decision time series for a representa-
tive set of 10 variables across all test ensembles. Within an individual test ensemble, the results were similar for all tested
variables, typically rejecting or passing in unison. This uniformity across variables is not surprising, given the strong cou-
pling by the governing equations. Note that this consistency remains true for the full set of 47 variables tested, except for soil
variables and the surface snow amount. Those exhibited a unique pattern, characterized by more persistent and occasionally
time-lagged rejections compared to their atmospheric counterparts.

The ID ensemble serves as a baseline and exhibits the fewest rejections, while the number of rejections increases progres-
sively through the modified diffusion ensembles, peaking in the C50 test. Meanwhile, our primary test ensemble, SP, exhibits
a low rejection rate for most variables comparable to the ID ensemble.

Temporal trends in the rejection rates were largely absent across all ensembles, although an initial spike in rejections occurred
within the first 10 to 20 days of the SP simulations. This initial spike is consistent with findings from Zeman and Schär (2022)
and can likely be attributed to the lower internal variability at the beginning of the simulations, making the test more sensitive to
differences. Figure 3 offers a time-averaged summary for all 47 variables, illustrating a gradually increasing rejection rate from
the ID ensemble to the modified diffusion tests. The ID results indicate that we can expect a false rejection rate of $10\%$ to $20\%$
for most variables. Such a high rate of false positives is likely the result of the high internal variability, probably undersampled
by only 100 members, of instantaneous or 12- or 24-hourly output fields at a grid-cell level for such long simulations. The
average rejection rate is almost always 2 to 5 percentage points higher for SP, and we see a steady increase from $\sim 40-60\%$
to $100\%$ for the modified diffusion experiments.

In the SP ensemble, six variables emerged as outliers: the four cloud cover variables, the deepest soil moisture metric, and
the height of the planetary boundary layer (HPBL). A closer examination at the individual grid point level reveals that the vast
majority of decisions for five out of six of the previously cited variables (all but HPBL) are related to a spurious interaction
between the Kolmogorov-Smirnov (KS) test and very steep distribution functions of bounded variables. When the steep slopes
of two distribution functions nearly coincide but are separated by a tiny distance on the order of machine precision ($\sim 10^{-7}$),
the KS test can incorrectly perceive this as a substantial vertical gap in the distributions, consequently leading to an unwarranted
rejection of the null hypothesis at that specific grid point. In summary, these near-near-ties can be considered artifacts of the
testing methodology rather than reflecting genuine model differences and are further explored in Appendix C. For variables





**Figure 2.** Monthly averaged test decisions for all five test ensembles and a representative subset of 10 variables out of 47.

showing this kind of behavior, a small rounding step (to the fifth or sixth decimal point) can be applied to the data from all ensembles before performing the testing to eliminate spurious rejections.

However, for the height of the planetary boundary layer (HPBL), the KS test artifact argument cannot be used, suggesting that the elevated rejection rate signals an underlying issue in the code. This highlights two key points. First, the identification potentially reveals code that is still sensitive to precision (i.e., a bug), thereby showcasing the effectiveness of our proposed testing method. Second, it is worth noting that HPBL is a purely diagnostic output variable in the COSMO model, so any imprecisions associated with it do not feed back into the subsequent model development. Nevertheless, the revealed differences in the HPBL between DP and SP necessitate further analysis and likely some adaptation of the corresponding source code.





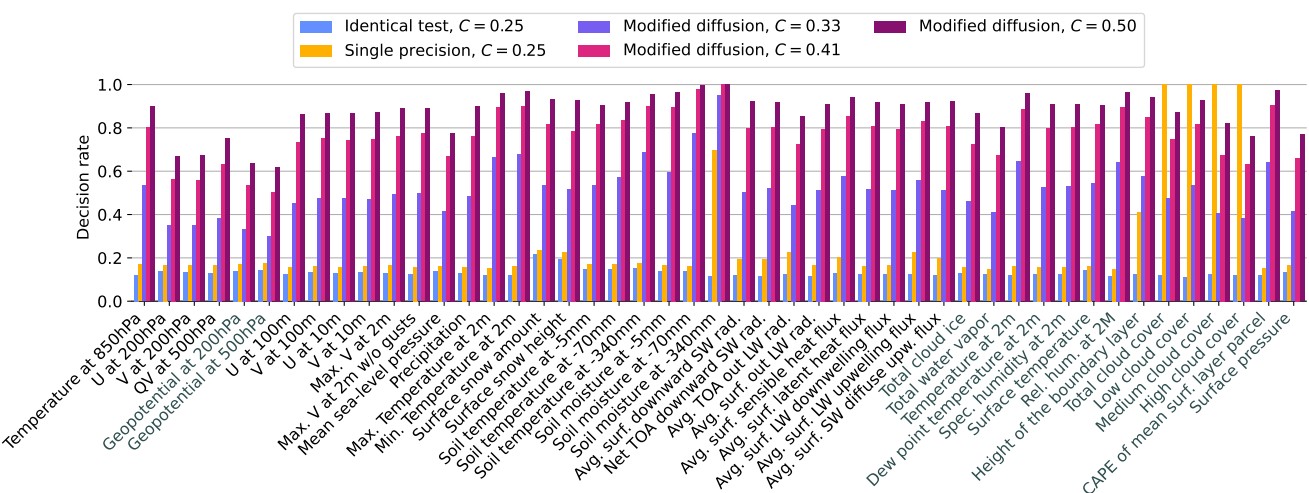

**Figure 3.** Time-averaged test decisions for all variables and all test ensembles considered in the study. Variables in black are output daily while those in gray are output 12-hourly.

Finally, it is worth noting that the radiation and soil model are run in double precision even in SP simulations, as running these components in their current state in SP causes too large errors. Surprisingly, variables linked to these modules still showed a slightly higher rejection rate in SP compared to ID, reflecting the coupling with the model components converted to SP.

## 4 Summary and discussion

Our results indicate that the methodology developed by Zeman and Schär (2022) is well-suited to climate simulations without requiring any modifications. It detects small changes in model parameters over long time scales while maintaining a reasonably low false-rejection rate. Notably, when applied to the output of SP climate simulations, the methodology demonstrates that these simulations are comparable in quality to DP simulations even after a 10-year period. The poor performance of the SP model for five variables is shown to be caused by technical artifacts rather than a significant difference between the SP and DP model results. However, a sixth variable, HPBL, also shows poorer performance than the rest, which cannot be attributed to the same artifacts but rather to a small bug in the code, which necessitates further analysis.

However, even this significant change of HPBL due to the use of SP comfortably lies within the variability of model results induced by the use of different model parameters as part of model tuning. Therefore, our results encourage exploiting the reduction of computational costs of around 30% obtained from reduced precision for regional climate simulations with COSMO. More generally, the results encourage the development towards the use of reduced precision also in other climate models.



To our knowledge, this work is the first to test the accuracy of reduced precision regional climate simulations on a comprehensive set of output variables using a state-of-the-art regional climate model.

    It is important to acknowledge that the tests conducted in this study were based on regional and not global climate simulations. In this type of simulation, the spread, or amount of internal variability, is inherently constrained by the lateral boundary conditions, which are exactly the same across the ensemble members. Nevertheless, the results of our sensitivity tests reveal

that even in the presence of boundary forcing, we are still able to detect differences within the domain caused by relatively minor changes to the model parameters. Nonetheless, it would be valuable to extend the application of this methodology to global simulations to evaluate its performance under those conditions as well. As the methodology works with any gridded field, it is directly applicable to the output of global climate models.

    Future users of the methodology developing their own SP implementation will not need to perform as long and costly

simulations as done in this work in order to test it. Simulations of a few days, as done in Zeman and Schär (2022), are long enough to test the dynamics and fast atmospheric processes. We therefore recommend running those as a first verification step. Additionally, we recommend running the model with a coarse resolution like the one used in this work or even coarser for the tests, as switching off parameterization modules also means that this part of the code will not be tested for differences, and to save computational resources.

Slower processes, including, in our case, soil and snow processes, but that may also include sea-ice, ocean or atmospheric chemistry processes in a fully-coupled global climate model, will need longer simulations. While a year is likely a safe choice for these processes, month-long simulations in both the cold and hot seasons would likely suffice to detect any major issues if the simulation costs are very high.

    We recommend comparing the results of the main test (here, SP) against an "anti-control" test produced by changing a model

parameter within its tuning range, as we did with the modified diffusion ensembles C33, C41, and C50. An idea that deserves to be explored further is the use of different parameters for different timescales or processes being focused on, like the soil hydraulic conductivity for soil processes. Furthermore, many of the results shown in this work could already be observed when working with ensembles of only 10 members each, but with typically less sensitivity. The effects of reduced ensemble and subsample sizes are explored more in Zeman and Schär (2022).

We hope that our methodology makes it easier for researchers to create and verify SP implementations of the models they are creating, which is a switch our results highly encourage. Our methodology can be used on relatively short and inexpensive simulations first before moving on to longer ones depending on the stage of development of the new implementation and on the physical processes being tested. It is good to keep in mind that our methodology provides per-variable, per-output-step results and does not directly give an overall statement about the model. It is up to the user to decide what amount of differences

between the new and the old implementation is acceptable, a decision that we believe is made easier when an anti-control ensemble is tested in parallel to the main test.



*Code availability.* The code for the data analysis presented in this work is available as (Banderier, 2023b). COSMO may be used for operational and for research applications by the members of the COSMO consortium. Moreover, within a license agreement, the COSMO model may be used for operational and research applications by other national (hydro-)meteorological services, universities, and research

institutes.

*Data availability.* The model is driven by boundary conditions extracted from the ERA-Interim dataset which is publically available. The input parameters for COSMO can be found in the repository where the rest of the code is located. The COSMO output data are over 10TB so they cannot be hosted online, but they can be made available upon request. Spatially averaged test results as well as final decisions of the methodology are published as (Banderier, 2023a).



## Appendix A: Comparison with the Benjamini-Hochberg procedure

A commonly used approach to determine the rejection of a global null hypothesis based on the rejections of local null hypotheses is the Benjamini-Hochberg procedure (Benjamini and Hochberg, 1995), which is also known as the false discovery rate (FDR) method. It has been extensively discussed by Wilks (2016). It is highly regarded within the scientific community and may offer a cost-effective alternative to our methodology, as it does not require a control ensemble or multiple subsampling steps. Comparing the results of the two methods, we observe that most output variables exhibit similar behavior (Fig. A1). However, certain variables, particularly those related to radiation, demonstrate a significantly higher or lower rejection rate in the single precision (SP) ensembles. Further examination reveals that these variables consistently exhibit clusters of unusually large rejections in specific regions, such as the western Sahara (not shown). We could not find an explanation for this behavior and further investigation, while certainly interesting, would be beyond the scope of this work. Confusingly, the radiation module operates using double precision floats in all models, and no other related variable displays the same behavior in the regions of high rejections for the radiation variables. The presence of these large spurious rejections leads to a higher number of rejected output steps under the FDR method compared to ours. Consequently, we believe that this method is overly sensitive to outlier grid points and may be less suitable for our specific case.

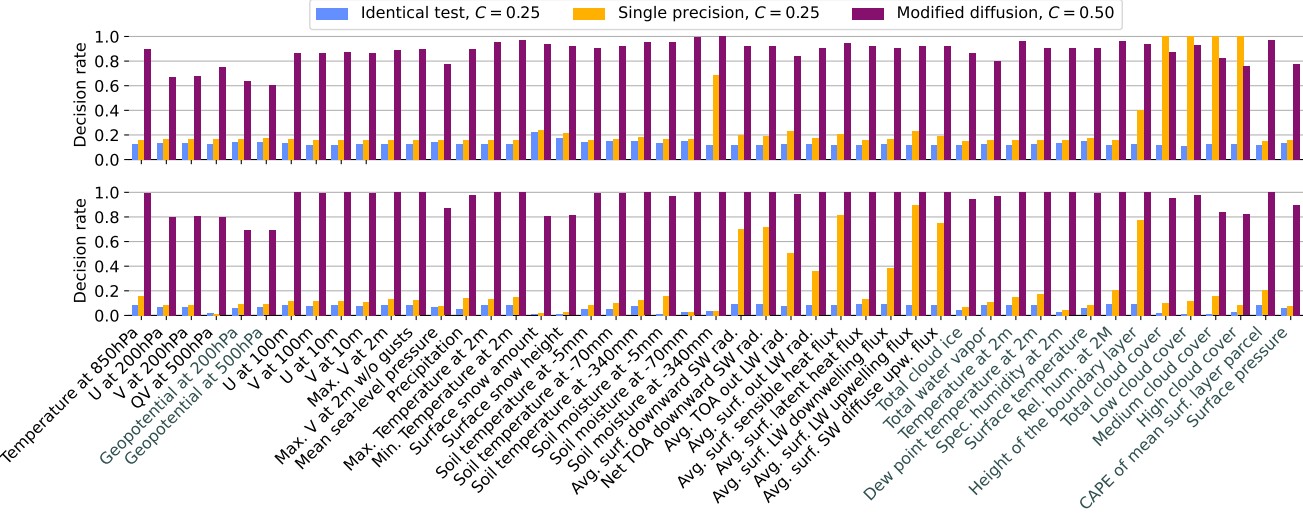

**Figure A1.** Comparison between our method (top) and the Benjamini-Hochberg procedure (bottom).

## Appendix B: Time coarsening

We now explore the effect of a coarser temporal resolution on the results presented in the main text by temporally averaging the data before applying our methodology. As a proof of concept, Fig. B1 shows the results for a single 2D field, temperature at





the 850 hPa level, and a single year at the end of our simulations for six decreasing temporal resolutions. As expected, the finer temporal structure of the results erodes with decreasing temporal resolution but the main features are there even if the data is averaged over the whole year. Interestingly, the sensitivity of the test for ID and SP reduces with decreasing temporal resolution.

Generally, we see less false positives with time averaging, which might be a result of the decreased internal variability coming from time averaging. However, time averaging also leads to a lower number of statistical tests performed for the same time period, which naturally leads to a lower number of false positives. Therefore, the relatively smooth rejection curves for the time-averaged results might be a bit misleading. For example, for the 1-year average, only one decision is made, naturally leading to a straight line in this diagram. Nevertheless, time coarsening may prove a valuable path forward if one is not interested in

the fine-grained details in the temporal distribution of the rejections or the model's representation of extreme events.

## Appendix C: Cloud cover and rounding problem

In the results section of the main text, we found six variables to have large decision rates for the SP ensemble, which seemed at odds with the fact that all other coupled variables (e.g., precipitation) showed low decision rates. To investigate this issue, we perform the first step of our methodology and apply the KS test to the ensemble distribution of randomly chosen grid points

and output time steps, and show the results for eight of them in Fig. C1. The dashed blue line represents the largest vertical distance between the empirical distribution and the $x$ position at which it is found. This maximum vertical distance is this test's statistic, which is then compared against a rejection threshold for the null hypothesis. Figure C1 shows that this maximum distance is always found at points where both distributions have a large positive derivative before reaching 1, and where both line seem to overlap. The issue is that the lines do not actually overlap but rather reach a different number. One reaches 1,

and the other reaches $1 - \varepsilon$, where $\varepsilon$ is small number $< 10^{-5}$. This is sufficient to create a large vertical distance between the two distributions at $x = 1 - \varepsilon$. We confirm this by rounding the variable to the fourth decimal for both ensembles, and observe in Fig. C2 that none of these days are rejected with this added rounding. A more comprehensive experiment shows that this mechanism completely explains the anomalously high rejection rate for these five variables (not shown).

*Author contributions.* HB and CZ designed the study. HB performed the ensemble simulations, wrote the code for the verification (based

on Zeman and Schär, 2022), and performed the verification and analysis of the model results with input from CZ and DL. All authors were involved in the discussion of the results. HB and CZ wrote the paper with strong contributions and review from all other co-authors.

*Competing interests.* The authors have declared that they do not have any competing interests.



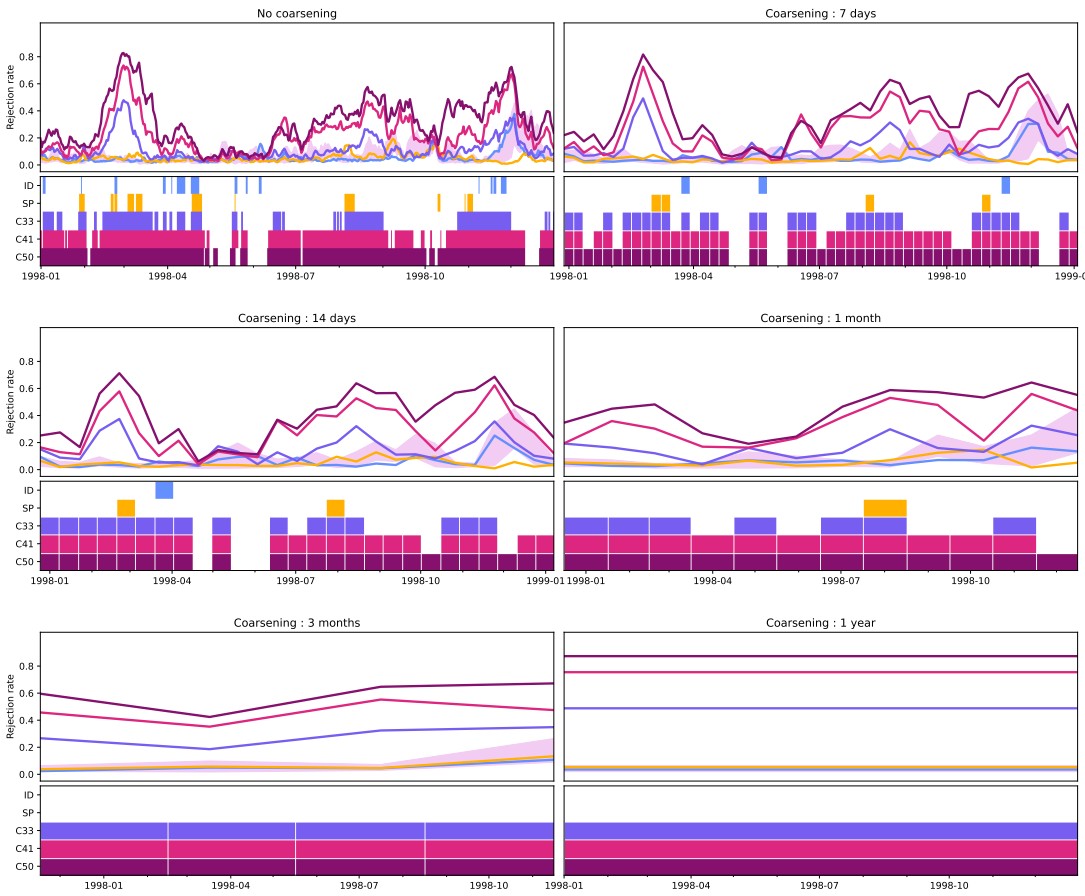

**Figure B1.** Rejection rate and associated decision (colored box = reject) for different magnitudes of time coarsening/averaging of temperature at 850 hPa for one year of output data.

*Acknowledgements.* We acknowledge PRACE for awarding computational resources for the COSMO simulations on Piz Daint at the Swiss National Supercomputing Centre (CSCS). We also acknowledge the Federal Office for Meteorology and Climatology MeteoSwiss, CSCS, and ETH Zurich for their contributions to the development of the GPU-accelerated version of COSMO with single-precision capability.






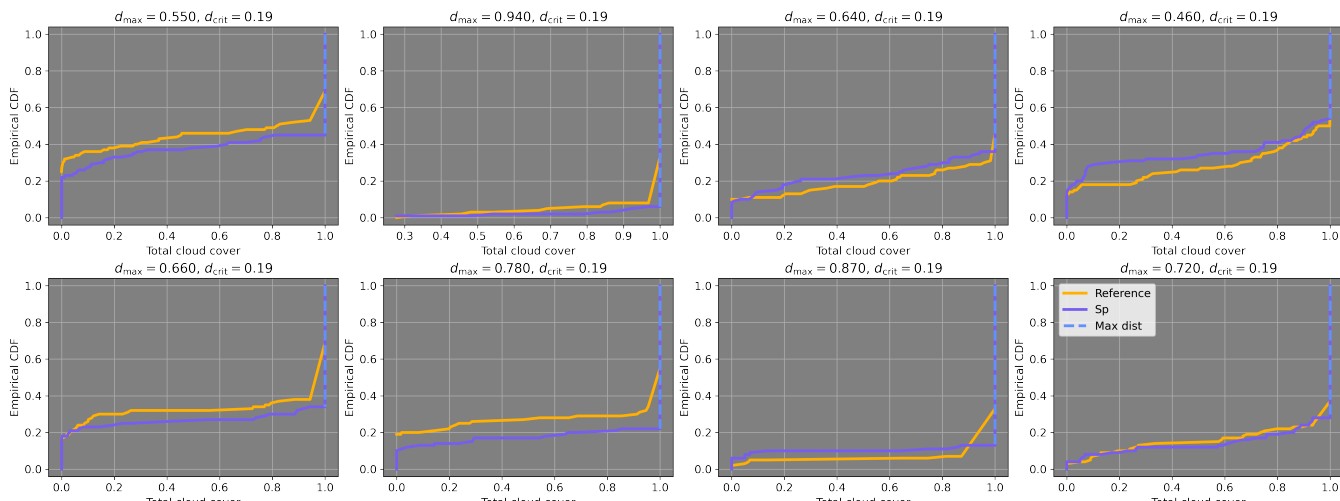

**Figure C1.** Example KS results for total cloud cover for eight random days without rounding. The gray background indicates that all these results lead to a local rejection.

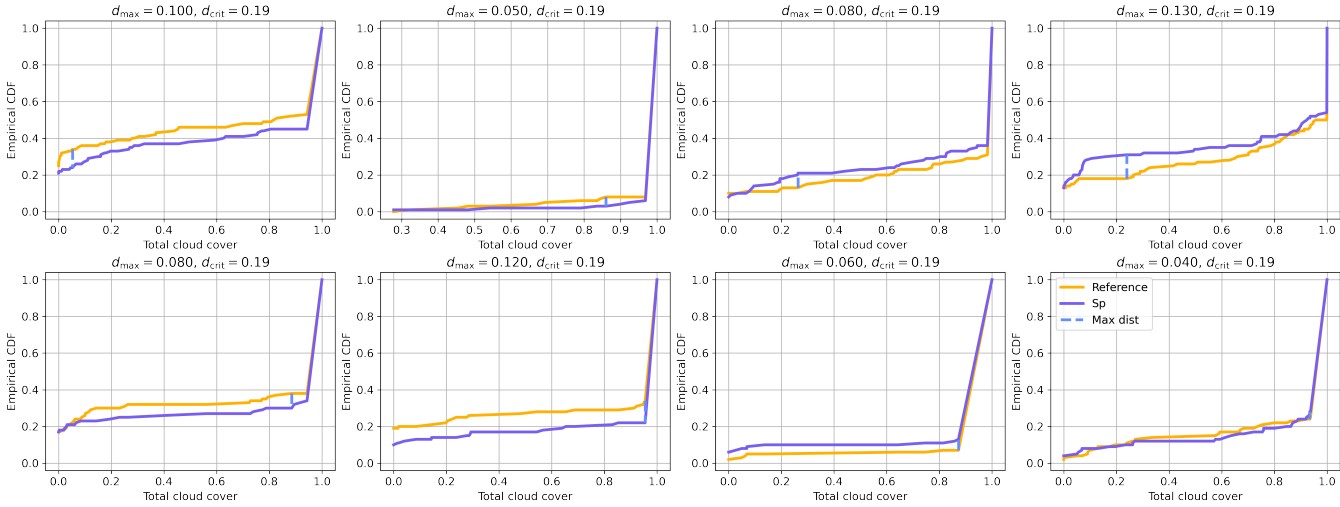

**Figure C2.** Example KS results for total cloud cover for the same eight days as Fig. C1 with rounding to the 4th decimal. Compared to Fig. C1, there are no local rejections when rounding is used.

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
