# Peer review of "Reduced floating-point precision in regional climate simulations: An ensemble-based statistical verification"

_EGUsphere, 2023_

## Author Comment (AC1)

The authors would like to thank the reviewers for their careful reading of our manuscript. We are grateful for their valuable comments and suggestions and address them in detail below.

*Referee 1*

*The manuscript is well written, concise, and represents an important contribution to the efforts of using reduced float precision in climate modelling.*
*The approach of ensemble based verification is interesting and thorough. The success of single precision is encouraging.*
*Additionally, I found the introduction to be a useful review of relevant literature.*
*The manuscript is in a good state for publication subject to some minor questions below.*
We thank the author for their positive comments. We hope our work will encourage more groups to consider the use of single precision simulations.

*- This is beyond the main scope of the paper, but it would be interesting to include some discussion on implications for 16-bit, stochastic rounding etc. especially since this is mentioned in the introduction.*
We thank the reviewer for their suggestion. We do not think that we would have any particularly striking new insight on these matters without additional experiments. We believe our verification scheme can be used to verify the results of such experiments.

*- Similarly, are there any plans for extensions to the work in 16-bit? I appreciate the hardware is not readily available; the papers you cite in the introduction use a half-precision emulator…*
This work focuses on the verification of a port of COSMO to SP that has been done in the past (Rüdisühli et al., 2014). While the verification can be applied to half precision simulation results without too many modifications, the actual port would likely require tremendous effort. As far as the authors know, and with the planned phase-out of COSMO in favor of ICON in many forecasting services, it is unlikely to happen for COSMO. We are not planning on pursuing this ourselves, through lack of either time or sufficient technical knowledge. Our hope is that the encouraging results in this work and all the literature we cite will encourage an in-depth port of ICON to single precision.

*- Some discussion/justification on the choice of KS over other metrics such as Wasserstein would be useful, especially since KS causes problems w.r.t the steep distribution functions of the bounded variables.*
The KS test was used because it can be made extremely fast, especially on a GPU, which was primordial for the amount of testing that needed to be done. It is also one of the tests used in Zeman and Schär 2022, to which this work is a successor, and thus some measure of continuity was deemed useful. Most other statistical tests that could be implemented to be reasonably fast would have suffered from the same problems (including all rank-based tests like MWU) or were not sensitive enough (student's $t$ test). As far as the authors understand, the Wasserstein distance does not have a closed form expression in general, and is instead computed as the solution to an optimization problem. This fact alone makes it surely too

expensive to compute at this scale. Still, it could be valuable to verify a small subset of our results using the Wasserstein distance.

*- The choice of the 95th percentile for the rejection of the global null hypothesis is reasonable, but I wonder how robust the results are to different percentile choices? The shape of the empirical distribution is surely relevant here which is somewhat lost by considering only the mean*

It is quite arbitrary, this is true. Sensitivity tests were done in the original paper but not again with this work. Some were performed again with the exact same setup as Appendix B, and the results are here. As expected, a higher quantile will lead to less rejections and a lower quantile will lead to more rejections for any ensemble. The key value to consider in the figure below is the difference in rejection rate between the identical ensemble, where every rejection is essentially a false positive, and the other ensembles. We see that the differences are relatively consistent throughout the higher quantiles, while a too low quantile makes the verification less sensitive due to a too high false positive rate.

[Figure]

Figure 1: With an similar setup to the manuscripts' figure B1: temperature at 850 hPa and the year 1998. For a range of 6 increasing decision quantiles, (each pair of panels), the top subpanel (identical in all 6) represents the mean rejection rates at each timestep for all 5 test

ensembles, and the (1-q)--q range of rejection rates for the control ensemble. The bottom subpanel represents the timestep-wise decision to reject or not this timestep based on the methodology and this particular decision quantile. The numbers on the right of the bottom subpanels are the proportion of rejected time steps in this example year with this example variable.

*- Last sentence of appendix C, some more details on the ""more comprehensive experiment" would be useful for reproducibility, even if the results are not shared*
It was a difficult choice exactly when to stop this discussion, as much more could be said about this topic. We updated the manuscript with an additional sentence to explain this experiment.

*- Figure 2: why are some variables labelled in grey, some in black? If it is the same reason as given in the caption of Figure 3 I would move the explanation earlier.*
It is the same reason. We thank you for your comment. The manuscript was updated to take it into account, and the explanation was added to Figure A1 as well.

*Referee 2*

*The authors present a statistical verification technique to compare two datasets whether are statistically from the same distribution or not. They apply this technique to an evaluation of single precision arithmetic used in COSMO (most model components) for 10-year climate simulations. The methodology is nicely illustrated (appreciation for Fig 1) and generally the paper is written for others to reproduce it, making it a useful manuscript for future projects. The method described is a statistical verifications between a (1) baseline model, (2) some change to it and (3) an "anti-control" ensemble used to quantify the expected deviation from the baseline due to uncertainties in parameterizations and tuning parameters. The authors therefore conclude that there is little reason to not use single precision in COSMO also for climate simulations, except for one purely diagnostic variable which the method identifies to suffer significantly from the use of single precision.*

*I generally recommend this paper to be published with minor corrections, see below for a list of points I created while reading the manuscript. Most of them are (strong) recommendations that you can object to though if you have a good reason for it (please explain). However, I also want to raise three "major" points that don't require major work, but that should be discussed in a paragraph or two as I feel this is currently missing from the text.*

*I enjoyed reading the manuscript, many thanks. It is generally well written and concise, illustrating a method and discussing some results in three figures without being overly complicated.*

We thank the author for their positive comments and their in-depth feedback. We hope to give a satisfactory response to all of them.

*Major points*
*1. Conditional distributions*

*Your method generally uses \*unconditional\* probability distributions. That means two adjacent grid points could co-vary in double precision but be independent with single precision even though both grid points follow an unchanged unconditional probability distribution. In an information theoretic sense, the mutual information between grid points could change even though the (unconditional) entropy is unchanged. As I see it, this could be a significant impact of precision (or any other change of the model) but your method would not detect it. A real world example might be precipitation which could occur in large patches (high mutual information) in the control but in smaller patches (lower mutual information) in the test ensemble. Clearly, analysing conditional probability distributions would explode the dimensionality of the problem, which is with 10TB already relatively large. Could you elaborate on this aspect in 2.1?*

This is a very astute point that is worth elaborating on. For an illustration of the example you are making, fig. 10 in https://gmd.copernicus.org/articles/14/4617/2021/ shows that increased diffusion (our choice for anti-control ensemble creation) increases correlation between grid points by creating larger patches of precipitation. In theory, a case may appear where this happens while, at the same time, the unconditional distributions for this timestep at all grid points are identical between the ensembles. Our methodology would then indeed not catch it and you are right that this is problematic. We think, however, that a case like this is too highly unlikely to pose any problem, since any change of this scale in the conditional probabilities is more than likely to also change the unconditional ones enough to be caught by our very sensitive methodology.

Another point in this direction comes from following the argument of Livezy and Chen (1983). If correlation between grid points increases, then the number of independent tests will decrease. A lower number of independent tests will for example make the probability of having a different rejection rate than in the reference higher. The distribution of rejection rates from subsampling might be narrower (for less correlation, i.e. higher number of independent tests) or wider (for more correlation, i.e. lower number of independent tests), but the mean of the rejection rate should remain around the same. As we compare this mean against the 0.95 quantile from the control rejection rate distribution, this should not affect the result.

This widening of rejection rate distributions can be very neatly observed for increasing diffusion coefficient, for all variables (not named but in the same order as figure 3 of the main text). Note that this is the rejection rate at the end of step b) of figure 1 and not the final rate of rejected timesteps at the end of step d).

[Figure]

Figure 2: Time mean of the daily member-wise standard deviation of rejection rates, for each variables and for the three ensembles named in the legend. This corresponds to the time mean spread of the distributions $f_{C-R}$ and $f_{C-T}$ in step c) of Figure 1 in the main text.

This theoretical point you raise is very interesting and worth mentioning in the main text. A short paragraph was added in that effect:

This methodology only tests for unconditional changes between ensembles. In an information theoretic sense, the mutual information between grid points could change between ensembles even though the (unconditional) distributions are unchanged. Our methodology would not catch this change, which is problematic. A methodology also testing for this would require quadratically more computational resources, which is beyond the computational resources available to most reasearch groups. This theoretical limitation probably has little effect in actuality, since a loss of mutual information would typically also come in enough unconditional differences to trigger a rejection.

*2. Rejection is binary*

*After reading the manuscript it is unclear to me what the effect is of the binary rejection, instead of having some error metric that would penalise larger deviations more. For example, if cloud cover has a rejection rate of n% because cloud cover is too high, then, as far as I understand it, it doesn't really matter whether in these situations cloud cover is 100% or 200%. But the latter is obviously not what you would tolerate a single precision simulation to output. I see the argument that you shouldn't penalise large deviations because maybe they don't matter more if they don't have kick-off effects causing other variables or grid points or timesteps to be rejected more frequently.*

This was a deliberate choice in the original paper that introduced this methodology. In this context, it was very valuable to give a simple yes-or-no decision per timestep, since it was introduced as, among other things, a continuous testing tool. We chose to keep the methodology as-is for clarity, since readers of this work can read the previous one for more details and motivation of the methodology.

Thanks to the high spatial correlation present everywhere, a large difference in magnitude at one grid point will likely also lead to many rejection in the neighboring grid point. Even if a field would theoretically have no spatial correlation at all, the often high correlation between different variables would make a large deviation also visible in other fields. The FDR method considers the magnitude of the rejection by looking at the p-values and it often leads to similar results as our method.

*3. Comparison to other methodologies*

*Reading Appendix A I'm left with the feeling that both methodologies have their outliers (for different reasons) and that an even better method would be to take the minimum rejection rate between both. Because then (if I see this right from Fig. A1) only HPBL would stick out as an anomaly which you also identify in the results. I see you have your arguments for your method over the Benjamini-Hochberg method but the discussion in Appendix C also shows that neither are actually methods robust to geophysical data distributions. I generally think the manuscript would gain a lot of strength if you incorporated the ideas from Appendix C directly into the main text to not leave the reader with a figure where you identify most variables that are outliers as being an artefact of the methodology. I mention in the minor comments that maybe converting the data to ranks, or maybe you can think of another way to deal with not-so-normal distributions. What you suggest to just round all data to 4 decimal points I think is one possibility (although I'd round in binary not in decimal) because your probability distributions should be well resolved by the numerical precision of your data. So whether you have 23 mantissa bits precision (in the data, not the compute) or 20 shouldn't have an impact on the rejection rate. I can see a method where you accept a rejection rate as robust when rounding it to n-1, n-2 or n-3 mantissa bits does not have an impact. But note that different variables have a different bitwise real information content. E.g. temperature or CO2 have much more information in the significand/mantissa than a variable that varies over orders of magnitude in the atmosphere, say, specific humidity (see Klöwer et al. 2021, https://doi.org/10.1038/s43588-021-00156-2). I'd love to see a version of the manuscript that does not require Appendix C to explain some artefacts.*

Thank you for this comment. You are right that neither of those methodologies are perfect in dealing with these kinds of issues. BH seems to suffer from not-so-normal distributions while our, in the configuration without rounding and with KS, suffers from the near-ties. Many of the different statistical tests (in the local testing step), like rank-based tests, would suffer from the same problems with near-ties, but I am sure that some others could solve them. KS was chosen in part due to how fast it could be made and GPU-parallelized, which was important for us seeing the amount of data that needed to be processed. In the end, we choose to stick with it and decimal point rounding.

Thanks to your comment, we investigated binary rounding, and found out more about the problematic values.

In the outputs of the double precision model, clct (the example we looked at) is never 0 but rather $5.4 \times 10^{-17}$, which is smaller than machine epsilon in double precision. It reaches 1

exactly however, but it might be a rounding between the double precision code and the single precision output. This value $\sim 10^{-17}$ is always the same and its exact value doesn't show any interesting pattern in its mantissa, which in other words means that binary rounding will have little effect on it.

In the outputs of the single precision model, clct reaches exactly 0 often, but never reaches 1 and instead *a wide range* of numbers near $1 - 10^{-6}$. Contrary to double precision's near-0 value, none of those near-one values are equal. The difference with one is this time larger than single precision's machine epsilon, and binary rounding has little effect on them.

We could not find the root cause of these confusing patterns and chose to just round everything at the 5$^{th}$ decimal point. It is an unsatisfactory solution, but it is the best one that does not involve a deep rework. In the updated version of the manuscript, all tests were redone with this rounding to make sure all variables are tested identically. Part of the imperfection feeling ties to your point about bitwise information, since a 5$^{th}$ decimal point rounding will have a different effect on a variable ranging between 0 and 1 than for one ranging 1 between 0 and $10^6$. The good thing is that this rounding has no effect on the results of all the other variables, and it "solves" the high rate of rejected timesteps for the 5 spurious spikes. HBL is also still anomalously high, as expected.

In the updated version of the manuscript, figures 2 and 3 of the main text were modified with results from a new round of testing that now includes rounding for all variables and all ensembles. Figure A1 however was kept as-is, to showcase the difference this rounding has made.

The updated version of the manuscript now makes a stronger point about SP being very close to ID and never worse than C33, and appendix C still justifies the use of rounding. We believe, thanks to your comment, the paper is now stronger.

*Minor points*

*Abstract*

*L11: """rejection rate" would benefit from a bit more explanation, rejected based on what? You of course elaborate on it in the text, but maybe just name the verification you use given this is the abstract? Maybe just "rejection rate, highlighting little statistical difference between the ..."*

Thank you. The manuscript was updated with the proposed change.

*L12: "negligible as masked by model uncertainty" maybe? To explain the meaning of your anti-control?*

Thank you. The manuscript was updated with the proposed change.

*Intro*

*L24: Maybe add memory and or data requirements?*

The manuscript was updated with the proposed change.

*L25: Or length of integration? Number of variables? Physical accuracy (e.g. more accurate parameterizations are often more costly), in the narrative of more accuracy with less precision?*

The manuscript was updated with the proposed change.

*L26: For most applications only float64 -> float32 is straightforward. 16-bit arithmetic often requires adjusted algorithms, and getting performance out is also not necessarily straightforward. You say this around L64 but maybe adjust the usage of "straightforward" here, or only refer to float64->float32.*

*L27: Remove "typically" and just state the important points of the IEEE-754 standard here.*

*L28: Note that this is the normal range, subnormals are smaller, please add to be precise.*

*L31: Note that around the number you mention the following float64 are representable*

> *296.45678912345664*
> *296.4567891234567*
> *296.45678912345676*

*Rounded to float32 the representable floats (round to nearest in the middle) are*

> *296.45676f0*
> *296.4568f0*
> *296.45682f0*

*While your point holds please choose a float64,float32 pair that's actually representable not "something like".*

Many thanks for this comment and the investigation. This was indeed a bit careless from our side. The manuscript was updated with the proposed change.

*L33: you mention discretization, model and boundary condition error, there's also initial condition error, maybe add?*

The manuscript was updated with the proposed change.

*L35: Maybe use "arithmetic intensity" to distinguish this concept from the use of "operational(ly)" in terms of operational, i.e. regularly scheduled numerical weather predictions?*

The manuscript was updated with the proposed change.

*L44: you switch between single precision and SP, I don't see the need to abbreviate but in any case be consistent?*

A consistency check was done. Thanks for pointing this out.

*L45: I would expect computing architecture or compiler settings to play a role too? Have you tested that too or is that less relevant given where and how COSMO is run? I'm not saying you should test that performance, but maybe just outline to the reader what could impact performance improvements.*

This is a good point and the numbers will likely also depend on the hardware and compiler settings. We did not test this, as we think this would be beyond the scope of this work. We have extended this part in the manuscript as follows:

"The reduction in computational costs from using COSMO with reduced precision is likely highly dependent on model configuration (model domain, domain decomposition, used parameterizations, etc.) as well as computer architecture and compiler settings. Previous studies with COSMO have shown the reduction in computational costs to be around 30% (Zeman and Schär, 2022) or 40% (Rüdisühli et al., 2014)."

*L97 and L98: the significanD or significanT bits.*

The manuscript was updated with the proposed change.

*L99: when first mentioning stochastic rounding, I'd provide a reference like https://doi.org/10.1098/rsos.211631*

Thank you. The manuscript was updated with the proposed change.

*L128: -to*

Thank you. The manuscript was updated with the proposed change.

*L150: I like these sentences summarising the implications of your rejection procedure. But it might be helpful to the reader to discuss, say, two cases, one where single precision causes a tiny bias globally, would this be rejected? As I see, not if that bias is masked by the variance of fC-R. And two, a case where single precision changes the climate in a small country but has no impacts in other regions of the world. Could these be added for clarification?*

Thank you for this comment. We have added the following sentences to the manuscript to discuss these two cases:

To exemplify the behavior of the procedure, we can think of two cases:

1. Suppose our test ensemble was constructed by a test model that mirrors the reference model but introduces a minuscule constant offset to an entire field. In that case, the likelihood of a local rejection would be marginally higher for each grid point.

Although this alteration may go unnoticed for numerous grid points, it would lead to an elevated mean of fC-T overall.

2. Our test ensemble might be generated by a model that induces a substantial change in a minuscule part of the domain but is otherwise identical. In this case, the grid points in this tiny part of the domain with a change will almost always lead to a local rejection, increasing the mean of fC-T.

Whether or not these changes would be detected or masked by the variance in fC-R heavily depends on the magnitude of the changes and other factors, such as the ensemble size and the length of the simulation. Previous results from Zeman and Schär (2022) show that the methodology is generally highly sensitive to changes, where, for example, an increase of the diffusion coefficient by only 0.001 could be detected.

*Figure 1: This is great!*

Thanks!

*L162: Call it multiplicative noise? There's an analogy here to the stochastically perturbed parameterization tendencies (SPPT) where the perturbations also take this form but you apply it to the actual variables, I assume R has some autocorrelation in space and time? Maybe irrelevant for your study though.*

This is much simpler than SPPT, there is nothing in what is done here that is as elaborate. R has no autocorrelation, or at least not by design or choice, it is simply a random number drawn (in theory) independently for all variables and grid points. The perturbation is applied exclusively to the initial conditions and no stochasticity is present at any subsequent timestep.

*L168: Maybe add a small table for the 7 ensembles?*

Good idea, thank you! This was added in the manuscript.

*L173: Can you not reduce diffusion because of numerical instability? If yes, maybe state why you're changing the coefficients only in one direction. Also while I see a change in diffusion as a reasonable control to test against, you could have also changed a physical parameterization (e.g. make convection stronger/weaker). Maybe elaborate more on your decision why you created the control as you did?*

We could just as well have reduced diffusion almost all the way to zero, as COSMO is run with little to no added diffusion in many cases (see for example Zeman et al., 2021). We focused on a gradual increase upwards rather than longer jumps in both directions because we thought it provided a more easily interpretable benchmark. Assessing whether or not a decrease of 0.25 would have a similarly large effect as an increase of the same amount was beyond the scope of what we are trying to show.

We could also have created the anti-control ensembles by changing parametrization, as was done in Zeman and Schär (2022) who, for example, tested changes in the SSO

parametrization with the same methodology. However, thi diffusion parameter has great properties that motivated our choice and that few other parameters have. It is always acting, as opposed to parametrization that require meeting criteria and thresholds to kick in, it is easy to understand and visualize, it can be gradually increased and decreased and finally it is a very common difference between models.

*Figure 2: Maybe state the precision on each panel? It's double everywhere where it's not single I guess, just for clarity.*

Good idea, this was changed in the manuscript.

*L187: Could you elaborate where rejections in the ID test come from? As I see it you only perturb the initial conditions so rejections are solely due to internal variability which however is small because of the identical boundary conditions leaving little room for the weather over Europe to evolve onto an independent trajectory? So this could be a storm away from the boundaries that is strong in some ensemble members but not in the other ensemble?*

Yes or any other synoptic scale pattern, or even a random bug/problem in enough ensembles at once. A sentence was added to the manuscript with this precision.

The internal variability is indeed rather small compared to, for example, ensembles that also contain continuously applied perturbations like SPPT. However, while a large variability might be desirable for forecasting, the methodology loses some sensitivity with a very high internal variability, which is exemplified by it being the most sensitive in the first few hours (see also Zeman and Schär, 2022).

*L193: Differences ... in the initial condition perturbations?*

After the first timestep, but the effect probably seems to come more from the very low variability in the reference/control test results than a high rejection in sp.

*L197: Given that some variables suffer from single precision as you outline, do you think this can have an impact on others being one average those 2-5% off? E.g. if cloud cover has a systematic bias with single precision this could introduce a bias on surface temperature (and consequently other variables) that's not large but enough to systematically cause those 2-5% higher rejection rate? Maybe a discussion of cross-variable impacts could be added?*

The cloud cover's high rate of rejected timesteps is mostly spurious, but it is not the case for the constant 2-5% increase for most of the other variables. The way we interpret it is that it comes from the interplay between small biases in many variables, rather than a large bias in one or two. If the cloud cover rejection signal was real, we would expect a much larger signal in directly covarying variables like precipitation, and the height of the planetary layer is a purely diagnostic variable so it also cannot be the culprit.

Our point is that, with enough time for all variables to interact with each other at all timescales, all the small biases have the opportunity to amplify each other, and they are still

much smaller than those created by a modest change in a commonly tweaked model parameter.

*L205: Use a rank-based test instead?*

Rank based tests would suffer from the same problem of "near-ties". With the examples of Appendix C but with Mann-Whitney U (MWU) test all the values at 1-eps would be inferior to the ones at 1, skewing the MWU statistic heavily and giving a similar result. This test was tried at the very beginning of the study since it was also used in Zeman and Schär (2022), but quickly abandoned also because of speed/scaling issues.

*L207: This sounds like also output precision (assuming you always output single?) is of relevance here. I think it makes sense to round all the data to something slightly less than single precision anyway. That way any clustering of data to identical values is at least the same across all simulations regardless of the precision used for computations.*

Yes, the only reason we didn't was to keep the methodology 100% similar to the original paper for clarity. As we discuss in more details in your third major point, we changed our mind since the paper is clearer with the rounding applied in the main text's results. The manuscript has been updated taking this into account, keeping appendix C and a few additional sentences as a justification for the modification, and figure A1 shows the unrounded results for comparison.

*L211: I find it difficult to think of every sensitivity to precision as a bug. There are algorithms that are stable only at high precision without them being coded up incorrectly. E.g. stagnation in large sums of small numbers due to insufficient precision can be overcome with a compensated summation but that comes at additional computational cost. In other situations you might be able to solve precision issues by computing the sum in reverse (possibly an easy fix that could be considered a "bug"). Maybe write "due to rounding errors in algorithms whether easy to fix (e.g. a bug) or not."*

Thank you for this comment. The manuscript was updated with the proposed change.

*L212: Could you mark those variables in Fig 3 somehow? For anyone repeating your analysis I find this an important concept to highlight that rounding errors from some variables cannot propagate to others which certainly helps in finding in which calculation precision is lost.*

Since we changed the methodology to add rounding and the spurious spikes in rejected timesteps disappeared, we did not do this.

*L215: noting -> reiterating given you already said this?*

The manuscript was updated with the proposed change.

*Fig 3: You write "Height of the boundary layer" but abbreviate it as HPBL, add the "planetary" or call it HBL for consistency? Also decision rate vs rejection rate?*

Thank you for pointing this out. We removed "planetary" and "P". it is now consistently "Height of the Boundary Layer (HBL)".

Rejections come at the end of step b) in figure 1, while the final decision rate, or rate of rejected timesteps, comes at the end of step d). We will try to make it more consistent in the updated version of the manuscript.

*L222: after -> during ?*

The manuscript was updated with the proposed change.

*L223: Could you not present a version of Fig 2 and 3 where these technical artefacts are somehow circumvented / the methodology adjusted? Most not-so-careful readers would probably look at those figures and conclude "single precision is bad for cloud or soil modelling so we shouldn't do this".*

As discussed earlier, this is what we do in the updated version of the manuscript. Thank you!

*L240ff: Just want to appreciate this list of recommendations that you give to readers, very helpful I believe!*

Thank you.

*L295: Temporal resolution usually comes with more constraints on compute and data storage. But would you recommend using time averages or time snapshots if both were available?*

Yes we also ran experiments with time averages (same as the grid coarsening) and also saw that the results hold with artificially decreased time resolution in the test.

*Fig C1: I find the grey background to highlight rejection a bit of an overkill, and it doesn't make the purple lines particularly readable, given all are rejected, just write this in the caption and make the background white again?*

Thank you. We made the background much lighter and hope the lines are now more readable. We still would like to keep it in a color other than white in order to make it clear that these CDFs will lead to a different outcome than the ones in white.